# Solid cyclooctatetraene-based triplet quencher demonstrating excellent suppression of singlet–triplet annihilation in optical and electrical excitation

Van T. N. Mai[1,2,9], Viqar Ahmad [1,3,9], Masashi Mamada [4,5,6], Toshiya Fukunaga[4,5], Atul Shukla [1,3], Jan Sobus[1,3], Gowri Krishnan[7], Evan G. Moore[2], Gunther G. Andersson [7], Chihaya Adachi[4,5,8 ✉], Ebinazar B. Namdas [1,3 ✉] & Shih-Chun Lo [1,2 ✉]

Triplet excitons have been identified as the major obstacle to the realisation of organic laser diodes, as accumulation of triplet excitons leads to significant losses under continuous wave (CW) operation and/or electrical excitation. Here, we report the design and synthesis of a solid-state organic triplet quencher, as well as in-depth studies of its dispersion into a solution processable bis-stilbene-based laser dye. By blending the laser dye with 20 wt% of the quencher, negligible effects on the ASE thresholds, but a complete suppression of singlet–triplet annihilation (STA) and a 20-fold increase in excited-state photostability of the laser dye under CW excitation, were achieved. We used small-area OLEDs (0.2 mm²) to demonstrate efficient STA suppression by the quencher in the nanosecond range, supported by simulations to provide insights into the observed STA quenching under electrical excitation. The results demonstrate excellent triplet quenching ability under both optical and electrical excitations in the nanosecond range, coupled with excellent solution processability.

[1] Centre for Organic Photonics & Electronics, The University of Queensland, Brisbane QLD 4072, Australia. [2] School of Chemistry and Molecular Biosciences, The University of Queensland, Brisbane QLD 4072, Australia. [3] School of Mathematics and Physics, The University of Queensland, Brisbane QLD 4072, Australia. [4] Center for Organic Photonics and Electronics Research (OPERA), Kyushu University, Nishi, Fukuoka 819-0395, Japan. [5] JST, ERATO, Adachi Molecular Exciton Engineering Project c/o Centre for Organic Photonics and Electronics Research (OPERA), Kyushu University, Nishi, Fukuoka 819-0395, Japan. [6] Academia-Industry Molecular Systems for Devices Research and Education Centre (AIMS), Kyushu University, Nishi, Fukuoka 819-0395, Japan. [7] Flinders Institute for Nanoscale Science and Technology, Flinders University, Sturt Road, Bedford Park, Adelaide SA 5042, Australia. [8] International Institute for Carbon Neutral Energy Research (WPI-I2CNER), Kyushu University, Nishi, Fukuoka 819-0395, Japan. [9] These authors contributed equally: Van T. N. Mai, Viqar Ahmad. ✉email: adachi@opera.kyushu-u.ac.jp; e.namdas@uq.edu.au; s.lo@uq.edu.au

The discovery of the first laser in 1960[1] has opened up a wide variety of applications, ranging from fundamental usages, such as scientific optical excitation and photolithography, to industrial laser cutting, drilling, military applications, medical imaging and surgery[2]. Compared to their inorganic counterparts, organic lasers offer many advantages such as compact size, high mechanical flexibility, high transparency and high wavelength tunability[3,4]. Moreover, among organic laser dyes, solution-processable dyes offer additional advantages of employing low-cost and large-area manufacturing techniques such as spin-coating or ink-jet printing in device fabrication.

So far, all organic lasers require a secondary excitation source such as gas lasers, inorganic solid-state lasers[5] or light-emitting diodes (LEDs)[6] for optical excitation. Direct electrical excitation of organic lasers, however, is significantly more challenging. While, very recently, Sandanayaka et al. demonstrated current-driven organic semiconductor laser diodes (OSLDs)[7], the current threshold is still extremely high with a value of around 1000 A cm$^{-2}$. This high current density inevitably creates a major issue for OSLDs, that is high accumulation of triplet excitons, leading to significant losses such as triplet absorption, singlet–triplet annihilation (STA)[8] and triplet-polaron annihilation (TPA). Given that in optical excitation there is only a small fraction of triplets converted from singlets via intersystem crossing, it is important to note that in electrical excitation, 75% of excitons generated are triplets according to spin statistics. This means that the large proportion of the non-emissive triplet excited-state together with its significantly long lifetime (often in microsecond range) leads to fast triplet accumulation in electrical excitation, compared to singlet excited-state (in ns range). These non-emissive triplet excited-states bring about above-mentioned losses, including the related stability issues[9]. As a result, lasing thresholds under electrical excitation are significantly higher than those under optical excitation[10]. Therefore, the ability to manage the non-emissive triplet states is crucial for the improvement of OSLDs.

Intelligent approaches to reducing the triplet excited-state population have been reported by using different triplet excited-state quenchers (TSQs)[9,11–16]. Among these TSQs, the most efficient quenchers are oxygen[17], anthracene[9,13] and cyclooctatetraene (COT)[14–16]. However, all of these TSQs have their respective limitations that render them incompatible for OSLDs. Specifically, due to the unique triplet ground state, molecular oxygen is easily converted into reactive singlet oxygen species, which is detrimental to the active organic semiconductor materials in the devices due to photo-oxidation and photodegradation, in addition to the undesired singlet quenching[18,19]. Anthracene and its derivatives are known to have relatively long triplet excited-state lifetimes ($\approx$20 ms)[20], where accumulation of triplets on the anthracene molecules is the key issue and source of another triplet accumulation[16]. In contrast, COT has been highlighted[16] as a more promising TSQ candidate thanks to its considerably shorter triplet excited-state lifetime (100 μs)[21] and low triplet energy without oxidising organic laser dyes[18,22,23]. Unfortunately, at ambient conditions, COT is a liquid (with a melting point of around −5 to −3 °C) and volatile. Hence, it has only been demonstrated for liquid-state organic dye lasers[23] as well as polyfluorene[18] but has not been compatible with thin-film devices and applications for real potential.

In this work, we designed and synthesised a solid-state organic TSQ based on COT, i.e., mCP-COT, in which mCP (i.e., N,N'-dicarbazolyl-3,5-benzene) is a common host with a high singlet and triplet energy, widely used in OLEDs. To maintain the individual electronic properties of mCP and COT, it is essential that a non-conjugated linker is employed. For mCP-COT to achieve a high solubility in common organic solvents for solution processing, without adversely affecting its thermal property, an n-hexyl linker was chosen for our initial study. We further used BSBCz-EH[24], which is a solution-processable version of state-of-the-art organic semiconductor BSBCz dye, to investigate its photophysical behaviours, ASE characteristics and photoluminescence (PL) stability. Notably, negligible effects on the film PLQYs and ASE thresholds of BSBCz-EH were found when blended with (equal or <20 wt%) mCP-COT. Under continuous-wave (CW) excitation, exceptionally efficient suppression of STA

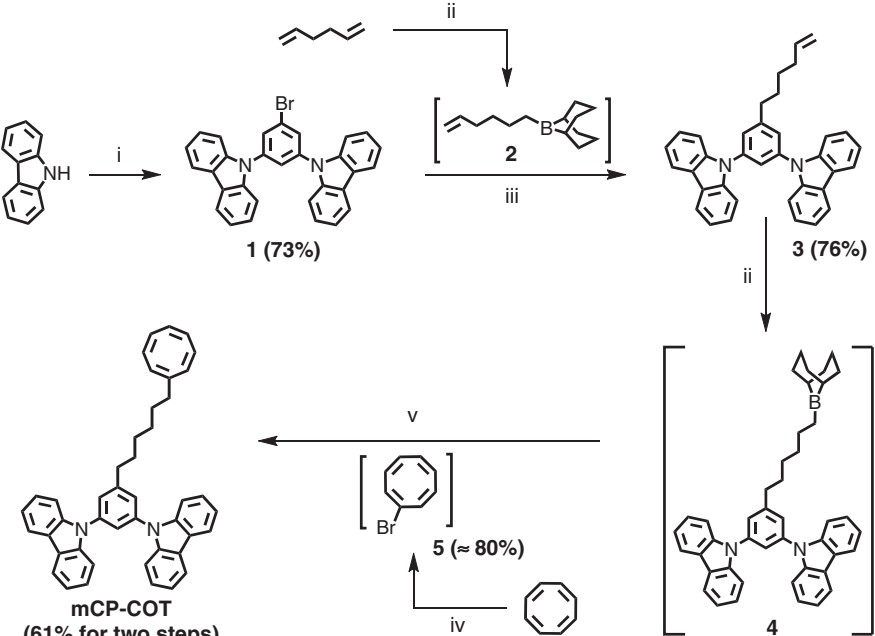

**Fig. 1 Synthetic route to mCP-COT.** (i**a**) $^t$BuOK, DMSO, Ar$_{(g)}$, 120 °C, 0.5 h, **b** 1-bromo-3,5-difluorobenzene, Ar$_{(g)}$, 140 °C, 0.5 h; (ii) 0.5 M 9-BBN in THF, Ar$_{(g)}$, r.t., 2.5–3.5 h; (iii) **2**, K$_2$CO$_3$, Pd(dppf)Cl$_2$.CH$_2$Cl$_2$, DMF, Ar$_{(g)}$, 60 °C, 16 h; (iv**a**) Br$_2$, CH$_2$Cl$_2$, Ar$_{(g)}$, −70 °C, 1 h, **b** $^t$BuOK, THF, Ar$_{(g)}$, −60 °C, 3 h; (v) **5**, K$_2$CO$_3$, Pd(dppf)Cl$_2$. CH$_2$Cl$_2$, DMF, H$_2$O, Ar$_{(g)}$, 55–60 °C, 17 h (DMSO: dimethyl sulfoxide; THF: tetrahydrofuran; dppf: bis(diphenylphosphino)ferrocene; DMF: dimethylformamide).

was achieved, attaining over 20-fold increase in PL stability compared to BSBCz-EH without mCP-COT. By using small-area OLED architectures, we demonstrated efficient STA suppression in ns range under electrical excitation. To provide more insights into the observed STA quenching of mCP-COT under electrical excitation, we conducted theoretical simulations. To the best of our knowledge, this is the first report of a non-emissive solid-state triplet quencher, exhibiting distinguished triplet quenching ability under both optical and electrical excitations, coupled with excellent solution processability. These results indicate the unique potential of mCP-COT towards the improvement of OSLDs.

## Results

**Material syntheses**. mCP-COT was synthesised as outlined in Fig. 1 where the synthetic details can be found in Supplementary Section 1. The first precursor **1** was prepared in a 73% yield from a commercially available 1-bromo-3,5-difluorobenzene with carbazole under nucleophilic aromatic substitution conditions. This was followed by a palladium-catalysed Suzuki-Miyaura cross-coupling reaction with a borane **2** to give **3** (76%), in which **2** was prepared by a hydroboration of 1,5-hexadiene with 9-borabicyclo [3.3.1]nonane (9-BBN)[25]. The similar hydroboration was further conducted for **3** to give borane **4**. To accomplish the synthesis, a mono-brominated COT was generated via a two-step reaction of COT to give **5** (≈80%). Finally, a similar palladium-catalysed Suzuki-Miyaura cross-coupling reaction of **4** with **5** was performed to give mCP-COT (61% for the two steps). mCP-COT was fully characterised as described in Supplementary Sections 1 and 2.

**Thermal and electrochemical properties**. Thermal properties of mCP-COT were studied by using thermogravimetric analysis (TGA) and differential scanning calorimetry (DSC). High thermal stability with a 5% weight loss temperature at 408 °C (Supplementary Fig. 1) was found, which is much higher than that (280 °C) of simple mCP[26]. DSC showed that mCP-COT has a glass transition temperature ($T_g$) at 55 °C (Supplementary Fig. 2), which is comparable to that (60 °C) of mCP[26]. In contrast to the parent liquid form of COT at ambient conditions, mCP-COT is a solid with a melting point ($T_m$) of 73 °C. Coupled with the high thermal properties, the results show our integration of COT with mCP enabling the material to become a solid at room temperature, which is of use for thin-film devices and applications. The electrochemical properties of mCP-COT were probed by using cyclic voltammetry (CV) (Supplementary Fig. 3) to show its redox behaviours arisen from the COT and carbazole species, respectively, which can be attributed to the non-conjugated linkage of the two electroactive species with different electronic properties. We also probed its neat-film ionisation potential (IP) and electron affinity by using ultra-violet photoelectron spectroscopy (UPS) and inverse photoemission spectroscopy (IPES), respectively, where the details can be found in the Supplementary Fig. 4.

**Photophysical properties and TD-DFT calculations**. Photophysical properties of mCP-COT were first investigated in toluene and compared to its parents, mCP and COT. The steady-state solution absorption and photoluminescence (PL) spectra as well as the absorption spectra of COT, mCP-COT and mCP were co-plotted in Fig. 2a with data summarised in Supplementary Table 1.

As can be seen from Fig. 2a, the solution absorption spectrum of mCP-COT is essentially the same with mCP (apart from a tiny red-shift), suggesting the transition is predominated by mCP unit whereas the small red-shifts are expected for such non-conjugated linkage[27,28]. Coupled with the fact that COT has low molar

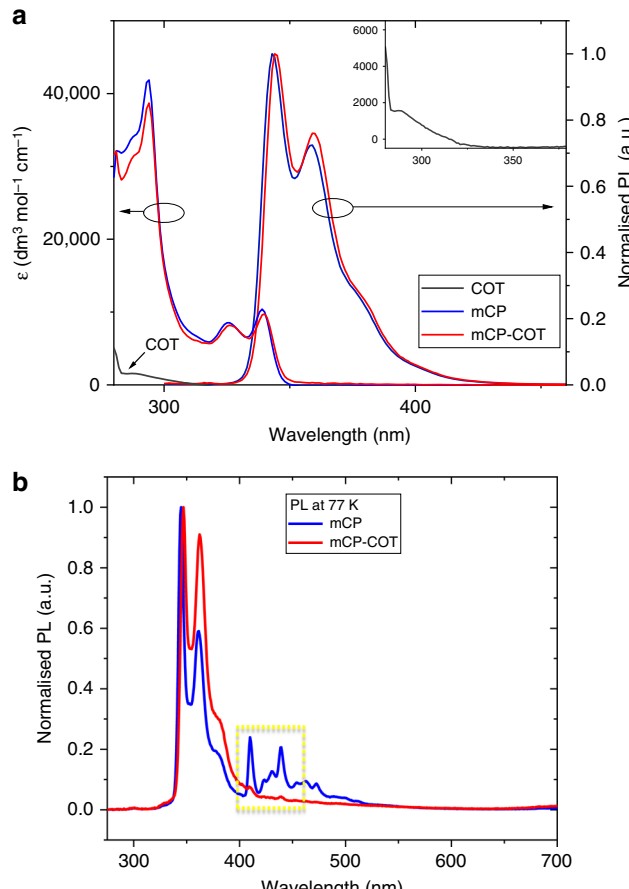

**Fig. 2 Room- and low-temperature solution absorption and PL Spectra. a** Solution absorption and normalised photoluminescence (PL, solid lines) spectra of COT*, mCP and mCP-COT in toluene (inset shows the weak COT absorption). Excitation wavelength = 290 nm. **b** Normalised photoluminescence of mCP (blue line) and mCP-COT (red line) at 77 K. At this low temperature, phosphorescence seen as peaks from 400 to 500 nm in mCP (yellow highlights) are not observed in mCP-COT, indicating efficient quenching of the triplet excited-state of the mCP moiety within mCP-COT. Excitation wavelength = 300 nm. *no PL was observed for COT.

extinction coefficient (with essential zero oscillator strength, see below TD-DFT calculations), the low energy absorption of mCP-COT peaked at 339 nm can be assigned as π→π* transition from the mCP moiety. While no PL was observed for COT, mCP-COT showed weak PL with a solution photoluminescence quantum yield (PLQY) of 6 ± 3% (in toluene). In essence, mCP-COT shares a similar PL spectrum to mCP with PL peaks at 344 and 343 nm, respectively, indicating the similar emission species due to the non-conjugation of the emissive mCP and non-emissive COT in the molecule. The considerable reduction in the solution PLQY of mCP-COT than that (43 ± 3%) of mCP can be attributed to the quenching by the COT moiety attached since it has lower singlet and triplet energies than the mCP unit—see the theoretical calculation section below. Such PL quenching is consistent with a much shorter excited-state lifetime of 1.5 ns for mCP-COT, as determined by time-correlated single-photon counting (TCSPC) with a 3rd order fitting (see Supplementary Fig. 5 and Supplementary Table 2), compared to that (5.3 ns in toluene) of mCP. Moreover, the phosphorescence of mCP-COT was significantly reduced (Fig. 2b), compared to mCP from our low-temperature PL measurements, indicating efficient triplet quenching by the COT component.

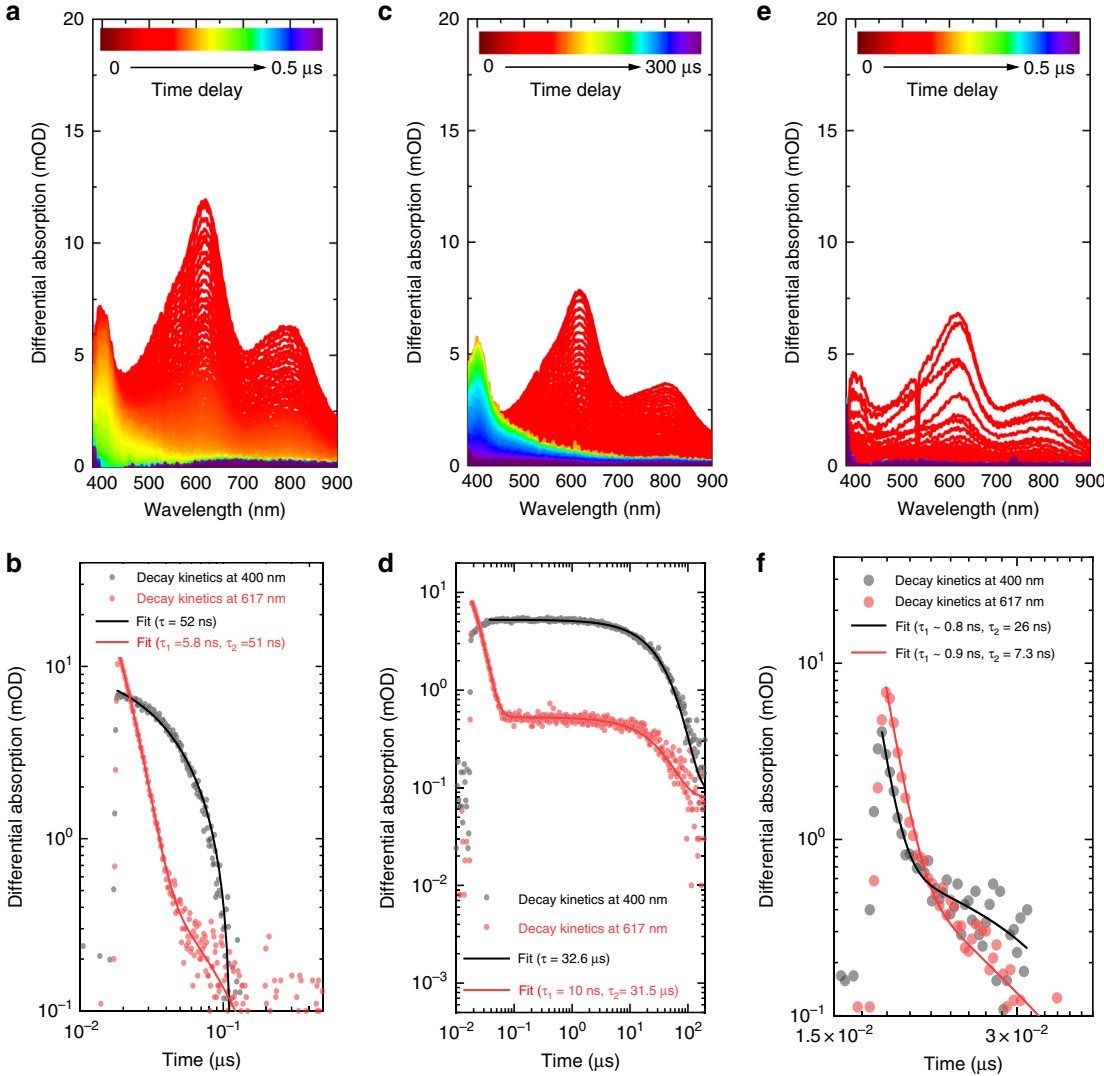

**Fig. 3 Transient absorption spectra and decay kinetics.** Decay kinetics were probed at 400 (black dots) and 617 nm (red dots), showing mCP in **a**, **b** ambient; **c**, **d** deoxygenated atmosphere, and mCP-COT in **e**, **f** deoxygenated atmosphere, respectively.

In order to gain further evidence of triplet quenching, we performed nanosecond transient absorption spectroscopy (TAS) for mCP and mCP-COT in acetonitrile. In ambient conditions, mCP showed long-lived excited-state absorption band with maximum at 400 nm (decay lifetime of 52 ns) and broad short-lived excited-state feature with maximum at 617 nm (bi-exponential lifetime of 5.8 and 51 ns) (Fig. 3a, b and Supplementary Fig. 6a). Herein, the decay kinetics of the short-lived absorption band (5.8 ns) were found to match closely with the singlet emission decay obtained from TCSPC (see Supplementary Table 2) measurements (5.3 ns) suggesting this transient absorption band arises due to the singlet excited-state absorption. In order to get further insights into the long-lived feature ($\tau \approx 50$ ns) we performed TAS for mCP under deoxygenated conditions. For deoxygenated solution (degassed using a freeze-pump-thaw method), the lifetime of the long-lived feature increased by more than two orders of magnitude ($\tau \approx 32$ μs) (Fig. 3c, d, and Supplementary Fig. 6b), suggesting this transient absorption band arises due to the triplet excited-states that were otherwise quenched by molecular oxygen under ambient conditions. In case of a deoxygenated mCP-COT solution, similar singlet and triplet excited-state absorption bands were observed. The decay lifetime of singlet excited-state absorption band was found to be 0.9 and

7.3 ns which is similar to the singlet emission lifetime obtained in the TCSPC measurements (Supplementary Table 2). Furthermore, triplet excited-state absorption decay of the mCP moiety at 400 nm was found to be significantly quenched ($\tau \approx 26$ ns) (Fig. 3e, f and Supplementary Fig. 6c). The shortened decay lifetime of mCP's triplet excited-state absorption in mCP-COT suggests ultrafast transfer of triplet excitons from the mCP unit to the COT moiety, though due to the low triplet energy level of COT, we could not observe the transient absorption band arising from the COT moiety alone. Supplementary Fig. 6d shows the normalised comparison of triplet excited-state absorption decay under ambient and degassed conditions.

As mentioned above, it is difficult to detect excited-state species of COT even though it is of importance to understand the energy levels of COT so to achieve good energy alignment with laser dyes. To get insights into these, we performed theoretical calculations and found that mCP-COT has two triplet energy levels, behaving like a non-vertical triplet quencher of COT. At ground state, the optimised structure of the COT moiety in mCP-COT was a non-planar tub-shaped conformation (Supplementary Fig. 7 and Supplementary Tables 3–5). It is important to note that the vertical excited-state energies for the first singlet ($S_1$-ver, 3.88 eV) and triplet ($T_1$-ver, 3.15 eV) of mCP are higher than

those (3.27 and 2.22 eV, respectively) of COT. Hence, the $S_2$-ver (3.89 eV) of mCP-COT corresponds to the $S_1$-ver of mCP, and the $S_1$-ver (3.32 eV) and $T_1$-ver (2.28 eV) of mCP-COT were nearly the same as those of COT, where the $S_2$-ver and $S_1$-ver transitions of mCP-COT are mainly populated over mCP moiety [HOMO → LUMO + 1 (57%)] and COT moiety [HOMO−2 → LUMO (100%)], respectively (Supplementary Table 4). However, the $S_1$-ver of COT is forbidden (with an oscillator strength of 0, Supplementary Table 4), which agrees with its absorption spectrum, appearing at shorter wavelength than mCP (Fig. 2a). On the other hand, the $S_1$-ver of mCP-COT appears to be an allowed transition because of the slight spreading of its MOs over the alkyl linker—see the MO distributions of its HOMO−2 and LUMO in Supplementary Fig. 7, which might be related to the decrease in PLQY for the doped films (vide infra). The structure relaxation with planarisation in the excited-states resulted in a significant decrease of the energies (see $S_1$-adi and $T_1$-adi for the adiabatic excitation), and the triplet quenching by COT is known to include non-vertical triplet energy transfer with conformational changes. Although the $S_1$-adi and $T_1$-adi energies were slightly different for COT and mCP-COT, it is considered that mCP-COT can also effectively quench triplet excitons with the same mechanism because of its low $T_1$-adi energy.

In contrast to simple mCP, mCP-COT was found to exhibit excellent solubility in common organic solvents, allowing for good-quality thin-film formation while using solution process, which is desirable to progress toward low-cost and room-temperature device fabrication by using techniques such as spin-coating or ink-jet printing. Consequently, a solution-processable version of the state-of-the-art organic semiconductor BSBCz dye[29,30], BSBCz-EH[24] (see the chemical structure in Supplementary Fig. 8a), was chosen as the active organic semiconductor laser dye for our mCP-COT triplet quenching studies.

Steady-state neat and blend-film absorption and PL spectra of BSBCz-EH with various mCP-COT blend concentrations (i.e., 1, 3, 5, 10, 20, 50 and 90 wt%) are shown in Supplementary Fig. 8. The absorption and PL maxima are nearly constant, suggesting no particular interactions in the excited-states of BSBCz-EH. We further determined PLQYs of these films and summarised the trend in Supplementary Fig. 9. High PLQY values of ≈70% were retained for the neat and blend films of BSBCz-EH with up to 20 wt% mCP-COT using excitation wavelength of 380 nm, where only BSBCz-EH was excited. Higher mCP-COT blend concentrations gradually decreased the blend-film PLQYs to 53 and 34% (for the 50 wt% and 90 wt% blends, respectively). Since blend films of BSBCz-EH in common hosts such as CBP showed high PLQYs[24], the decrease of blend-film PLQYs of BSBCz-EH with mCP-COT can, therefore, be ascribed to the quenching by the COT moiety. However, the quenching seems to be not efficient at moderately high PLQYs even with high COT concentrations (i.e., BSBCz-EH:COT = 1:17 mol/mol in the blend film of BSBCz-EH with 90 wt% mCP-COT), probably due to complicated energy transfer processes. It is interesting to note that the decrease in PLQYs with increasing mCP-COT concentration is dependent on the singlet/triplet energy of the emitter. Thus, the higher the singlet/triplet energy of the emitter, the sharper the decrease in PLQYs with increasing mCP-COT concentrations as shown in Supplementary Fig. 9 for CBP ($S_1 = 3.5$ eV, most quenching), BSBCz-EH ($S_1 = 2.9$ eV) and BSBCz-CN-EH ($S_1 = 2.6$ eV, least quenching).

**Triplet quenching studies under optical excitation**. First, the PL transient responses of encapsulated BSBCz-EH neat and blend films with different mCP-COT blend concentrations (5, 10 and 20 wt%) as well as a blend film with 20 wt% mCP were

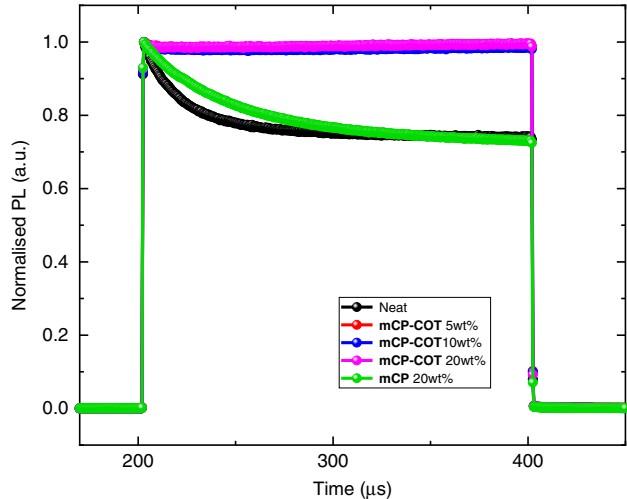

**Fig. 4 Transient PL of BSBCz-EH neat and blend films.** Characteristics of encapsulated BSBCz-EH neat film (black line) and blend films with different mCP-COT blending concentrations [5 wt% (red), 10 wt% (blue), and 20 wt% (pink)], which can be compared with its blend film with 20 wt% in mCP (green). Excitation wavelength = 355 nm; laser beam excitation power = 2.65 mW; pulse width = 200 μs and pulse interval = 10 ms.

investigated. These spin-coated thin films were excited at 355 nm using a circular beam with a diameter of 200 μm. The excitation power, pulse width and pulse interval of the laser beam were 2.65 mW, 200 μs, and 10 ms, respectively. All obtained PL spectra were normalised at their initial PL intensities as shown in Fig. 4. A significant decrease in transient PL intensity under CW excitation for BSBCz-EH neat and the blend films (with 20 wt% mCP) can be clearly observed. The initial PL intensities of both films were reduced by ≈25% under a pulse width of 200 μs. These reductions can be attributed to quenching of the emissive singlet excitons [i.e., singlet–triplet annihilation (STA)] due to accumulation of the long-lived triplet excited-state generated via intersystem crossing. In contrast, almost completely no reduction in the PL intensity for the blend films of mCP-COT was observed, regardless of blend concentrations, indicating absolute suppression of STA, which in turns advocates the exceptional triplet quenching ability of mCP-COT. While partial STA suppression under CW excitation had previously been reported using an anthracene derivative (i.e., ADN)[31], we noted that these films were vacuum-deposited and that significantly higher blending concentration of the triplet quencher ADN was required while the extent of STA reductions were not as effective. A comparative analysis of the triplet quenching performance of ADN and mCP-COT was conducted to show the relative drop in the initial STA in these two systems (Supplementary Fig. 10). These results support the superior triplet management properties of mCP-COT, since at the 10 wt% concentration mCP-COT removes over 98% STA present originally in the neat system, while the same concentration of ADN results in ≈25% reduction of STA (Supplementary Fig. 10). TCSPC PL study for BSBCz-EH also showed fast PL decay and high singlet decay rate ($k_r$) (see Supplementary Fig. 11 and Supplementary Table 6). To the best of our knowledge, such superior performance of a solid-state triplet quencher is unprecedented.

Given that triplet excited-states have been known as a prominent source responsible for photodegradation of BSBCz dyes[9], we investigated the effect of the triplet quencher mCP-COT on the photostability of BSBCz-EH. An encapsulated BSBCz-EH neat film, a blend film with 20 wt% mCP, and a blend

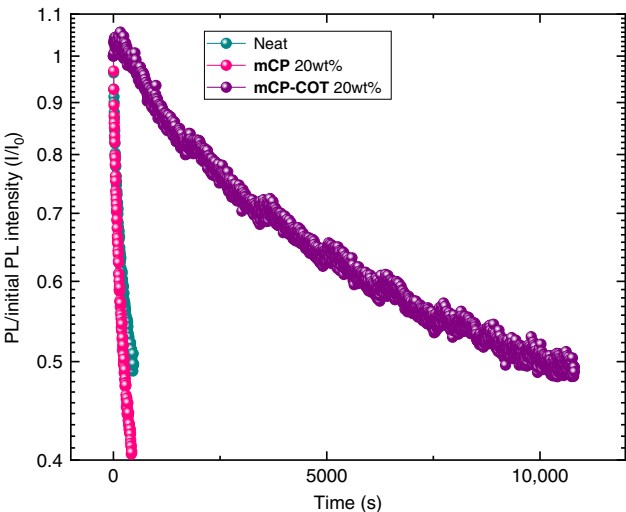

**Fig. 5 Photostability of BSBCz-EH in neat and blend films.** PL intensity/initial PL intensity ($I/I_0$) of a BSBCz-EH neat film (green line), a blend film with 20 wt% mCP (pink line), and a blend film with 20 wt% mCP-COT (violet line) measured under CW photoexcitation with a power of 200 mW cm$^{-2}$ at 405 nm. Excitation area = 2.5 mm × 2.5 mm circle.

film with 20 wt% mCP-COT were excited under CW photoexcitation with a power of 200 mW cm$^{-2}$ at 405 nm. The PL intensity/initial PL intensity ($I/I_0$) curves of these films are shown in Fig. 5. The times required for the corresponding films to reach half of their initial PL intensity (i.e., $I/I_0 = 0.5$) were found to be 428, 252 and 10,000 s, respectively. This shows that a 20 wt% additive of mCP-COT resulted in more than 20-fold increase in the sustenance of the PL duration, compared to the neat films, indicating the excellent triplet quenching ability of mCP-COT.

Next, ASE thresholds of neat and blend films of BSBCz-EH with mCP-COT at 5, 10 and 20 wt% blending concentrations were measured (Supplementary Figs. 12, 13). The ASE thresholds of the films blended with mCP-COT varied between 1.37 and 1.56 μJ cm$^{-2}$, which are comparable to that (1.32 μJ cm$^{-2}$) of a BSBCz-EH neat film measured under the same experimental conditions (Supplementary Fig. 13). The results have demonstrated that the use of mCP-COT as a triplet-state quencher additive has a negligible effect on the ASE properties of BSBCz-EH dye.

**Triplet quenching studies under electrical excitation.** While multiple studies on STA quenching under optical excitation have been reported[11,16,31], to the best of our knowledge, no report can be found on STA quenching study with electrical excitation in ns pulse width range, although there are some studies conducted in the microsecond regime but without revealing deep insights into the initial transients of the OLED devices[13,32]. To realise STA quenching, OLEDs with a small area are required (≪1 mm$^2$) to limit the resistor-capacitor (RC) constant of the devices, as small RC constant allows rapid response (in ns) of the devices to be detected. In contrast, a device with pixel area of several mm$^2$, where the EL signal evolves slower in time due to larger capacitance, results in loss of STA quenching visibility in the EL signal. We used BSBCz-EH with mCP-COT blend to realise OLEDs for testing STA quenching and compared with a neat BSBCz-EH OLED device.

Accordingly, the structure of small-area OLEDs studied was ITO (100 nm)/PEDOT:PSS (30 nm)/BSBCz-EH (neat or with 2 wt% mCP-COT) (60 nm)/TPBi (40 nm)/LiF (1 nm)/Al (100 nm), where ITO is Indium tin oxide, PEDOT:PSS is poly(3,4-ethylenedioxythiophene)-poly(styrenesulfonate) and TPBi is 1,3,5-tris(2-N-phenylbenzimidazolyl)benzene. The device area of OLEDs was 0.2 mm$^2$. To achieve high current densities and high luminance, a fundamental requirement for injection lasing, the small-area OLEDs were subjected to pulse widths of 100 ns at voltages varying from 20 to 100 V, where the nanosecond pulse widths also enable limitation of the Joule heating.

Once the nanosecond pulse was applied to the small-area OLEDs, EL signals were generated in the organic emissive layer with a delay resulting from the time needed for holes and electrons to form excitons. Singlet excitons are short-lived and usually have a lifetime of few ns for fluorescent emitters (e.g., 1.4 and 1.5 ns for neat BSBCz-EH and blend BSBCZ-EH with 1 wt% COT, respectively), whereas triplets are slower to reach maximum concentration but are generally orders of magnitude higher in density than singlets due to the longer lifetimes. These non-radiative triplets annihilate singlets causing STA, resulting in higher energy triplets and charge carriers[33]. The evidence of STA can be seen in an EL waveform as a reduction in EL intensity after the initial EL peak within tens of nanoseconds. This drop in intensity depends on the STA rate after which EL intensity achieves a steady state. The higher the STA, the more reduction in EL intensity compared to its peak value.

Figure 6a shows EL response of the neat and blend (with 2 wt% mCP-COT) BSBCz-EH based OLEDs to a 100 ns pulse input, where a considerable reduction in EL intensity can be seen in case of the neat device under the same current density of 50 A cm$^{-2}$. Figure 6b shows normalised EL intensities of the neat and blend OLEDs where a substantial STA can be seen for the neat device. The reduction in intensity after the initial peak was around 25% for the same current. Comparing Fig. 6a, b, it is evident that mCP-COT has aided in the reduction of STA. Figure 6c, d shows plots of EQE and brightness *versus* current density, presenting a similar order of magnitude improvement in EQEs and brightness.

In order to confirm STA quenching by mCP-COT in the blend films, rate equations for polaron, singlet and triplet generation were simulated in MATLAB® and the STA rate along with other annihilation rates was extracted from the programme. Simulation of neat and blend device EL characteristics from rate equations (see Supplementary Eq. 1) suggests an STA rate ($k_{STA}$) of $4.3 \times 10^{-8}$ cm$^3$ s$^{-1}$ for the neat OLEDs. For blend OLEDs, $k_{STA}$ was kept the same and a term, $k_{mCP-COT}$, was introduced in the triplet equation depicting contribution of mCP-COT towards rapid triplet depopulation. $k_{mCP-COT}$ was extracted to be $1 \times 10^{10}$ s$^{-1}$. Supplementary Fig. 14a, b shows the result of rate equation fitting for the EL response of the neat and blend devices, respectively. It must be noted that the plotted singlet density for both neat and blend devices is for the same current (50 A cm$^{-2}$) going through both devices. However, singlet density for the blend can be seen as being around eight times the singlet density in neat device (an indication of more STA quenching in neat device). The results of reduced STA quenching indicate the triplet quencher mCP-COT is efficient for the fast triplet decay. Supplementary Fig. 14c gives evidence of triplet populations extracted from neat and blend devices. The triplet population obtained for neat devices is almost 30 times more than that of the blend.

DC characteristics of the same OLEDs can be seen in Supplementary Fig. 15a. The blend OLEDs of BSBCz-EH outperform the neat device with EQE reaching close to its theoretical limit of ≈4% (calculated based on PLQYs of ≈70% for a fluorescent dye and out-coupling factor of 0.2). The *J–V* characteristics are also very similar for neat and blend OLEDs as

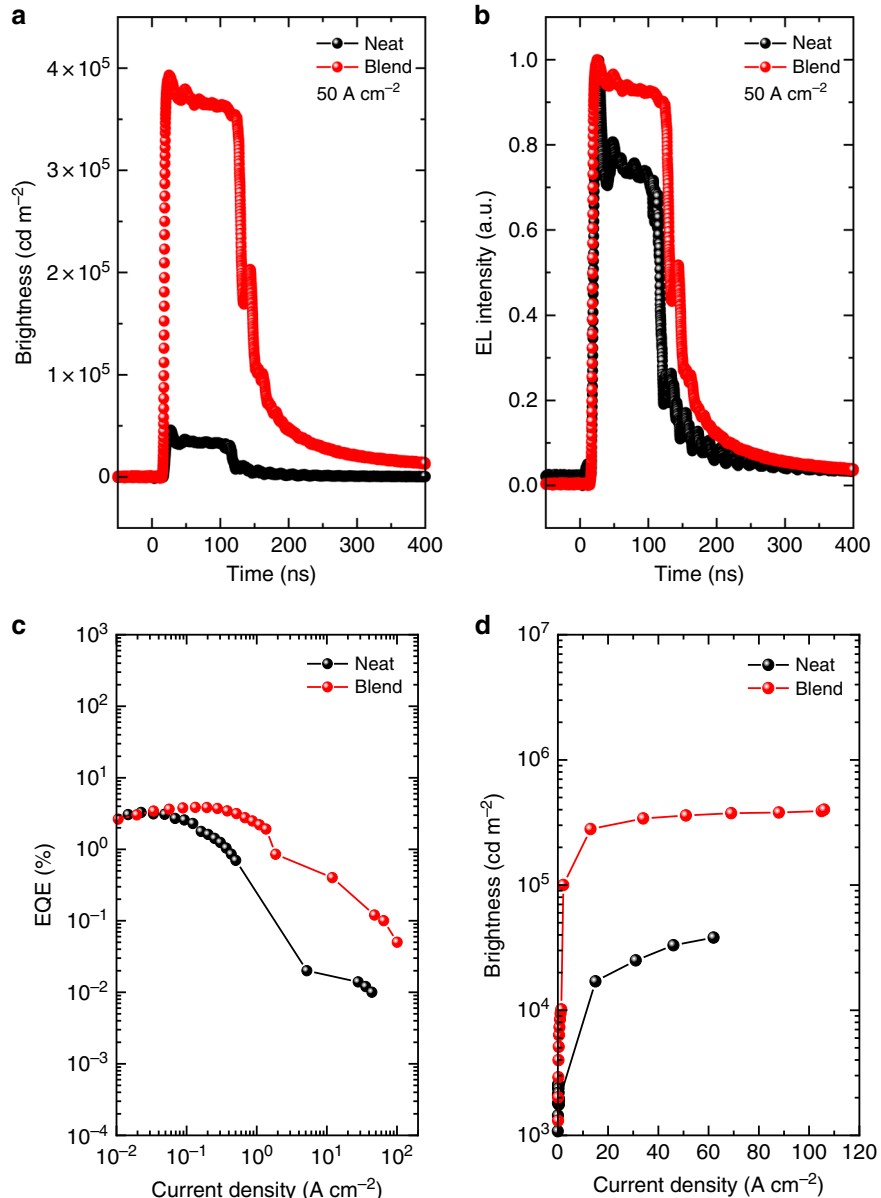

**Fig. 6 Brightness and EQE comparison of neat (black) and mCP-COT blend (red) small-area OLEDs for pulse inputs. a** A comparison of EL intensity of the neat and blend BSBCz-EH OLEDs at a current density of 50 A cm$^{-2}$. **b** Normalised EL intensities, showing a significant STA in the neat device and substantial reduction of the same in blend. **c** EQE versus current density for neat and blend devices at 50 A cm$^{-2}$. **d** Brightness versus current density plot for neat and blend devices. OLED area = 0.2 mm$^2$; pulse widths = 100 ns.

shown in Supplementary Fig. 15b, depicting no change in electrical behaviour with the addition of mCP-COT in the operating voltage region (>4 V).

## Discussion

We have successfully designed and synthesised an efficient solid-state organic triplet quencher mCP-COT that exhibits excellent solution processability. Photophysical, thermal and electronic properties of the compound were reported, showing the integration of COT onto mCP enables mCP-COT to be a solid at room temperature with a high decomposition temperature, useful for thin-film devices and applications. The triplet quenching ability of mCP-COT as an additive was investigated using the solution-processable dye BSBCz-EH under both optical and electrical excitations. Under CW photoexcitation, even small blending concentrations of mCP-COT

(equal or less than 20 wt%) showed unprecedented STA suppression. Moreover, over 20-fold improvement in PL stability of BSBCz-EH was found by blending with mCP-COT. Notably, negligible impact on the ASE characteristics of the dye BSBCz-EH was observed as comparable ASE threshold values were measured for the blend films compared to the neat films of BSBCz-EH. Remarkably, the dynamics of STA suppression under electrical excitation in ns regime were demonstrated by employing small-area OLED structures. Compared to a BSBCz-EH neat film, a 2 wt% blend of mCP-COT resulted in a substantial reduction of STA observed in the EL-current density curve, coupled with an increase in EL intensity and brightness, comparable EQE maximum, as well as a reduction in EQE roll-off. In addition, theoretical simulations were conducted to provide insights into the observed STA quenching of mCP-COT under electrical excitation. This is the first report of a non-emissive solid-state triplet quencher, exhibiting excellent

triplet quenching ability under both optical and electrical excitation, coupled with excellent solution processability.

## Methods

**General**. All commercial reagents and chemicals were used as received unless otherwise noted. Tetrahydrofuran (THF) and *N,N*-dimethylformamide (DMF) were dried using a vacuum-argon solvent purification system before use. Dimethyl sulfoxide (DMSO) was stirred overnight with calcium hydride (3% w/v), distilled and stored in activated 4 Å molecular sieves under argon. Dichloromethane was dried with calcium hydride (3% w/v) overnight and freshly distilled prior to use.

**Material syntheses and characterisations**. Detailed synthetic protocols and characterisation data can be found in Supplementary Section S1, while $^1$H (500 MHz, CDCl$_3$) and $^{13}$C NMR (125 MHz, CDCl$_3$) spectra of the compounds can be found in Supplementary Section 4.

**Photophysical measurements**. All substrates were cleaned by sonication in acetone and isopropanol, followed by UV-ozone treatment. Thin films were fabricated by spin-coating from a chlorobenzene solution of 1.0 wt% fluorescent dyes and mCP-COT at 1000 rpm for 60 s on clean quartz substrates. UV–Vis absorption spectra were measured using Perkin-Elmer Lambda 950-PKA UV–vis spectro-photometer. Photoluminescence (PL) spectra were measured using Horiba Jobin Yvon FluoromMax-4. PLQYs were measured using an absolute PLQY measure-ment system (Hamamatsu Photonics Quantaurus-QY C11347-01). The measure-ment error for the obtained QY values on this instrument was ±3%. Transient photoluminescence decay was recorded on a Hamamatsu Photonics Quantaurus-Tau C11367-03. Phosphorescence spectra were recorded on the Hamamatsu Photonics PMA-12 with the LED excitation at 300 nm (Thorlabs M300L4).

**Computational studies**. The computations were mainly performed using the computer facilities at the Research Institute for Information Technology, Kyushu University. Molecular orbital calculations were performed using the programs Gaussian 16. The geometries for the ground state were optimized at the B3LYP/6-31+G(d,p) level. The presence of energy minima was confirmed by the absence of imaginary modes (no imaginary frequencies). The time-dependent density func-tional theory (TD-DFT) calculations were conducted at the B3LYP/6-31+G(d,p) level. To numerically achieve accurate values, we have used a fine grid. The sol-vation effect in a solvent was considered by using the polarizable continuum model (PCM). Since the optimization at the excited-states for COT and mCP-COT using TD-DFT resulted in convergence failures, TD-B3LYP/6-31+G(d,p)//CIS/6-31+G(d,p) was used, while the optimization at the excited-states of mCP was conducted using TD-B3LYP/6-31+G(d,p). The optimized structures for the triplet excited-state were also calculated using UB3LYP/6-31+G(d,p).

**ASE measurements**. Thin films were prepared by spin-coating from 1.5 wt% chloroform solution at 1000 rpm for 60 s on non-fluorescent glass substrates (MATSUNAMI slide glass S0313). The substrates were cut to use the centre of the substrate with smooth flat surface. ASE properties of the thin films were char-acterised by optically pumping with a randomly polarised nitrogen gas laser (KEN2020, Usho Optical Systems Co., Ltd.) at an excitation wavelength of 337 nm with a 0.8 ns pulse (operating frequency of 10 Hz). The input laser beam was focused into a stripe with dimensions of 0.6 cm × 0.12 cm using a cylindrical lens. Neutral density filters were used to adjust excitation intensity. ASE measurements were performed under a nitrogen atmosphere. Output light emission from the edge of the sample was collected into an optical fibre connected to a spectrometer (Hamamatsu Photonics PMA-12). ASE thresholds were identified from the plot of output versus input intensity.

**Transient PL measurements**. Thin films were prepared using the same condition as those for photophysical measurements on non-fluorescent glass substrates, which were encapsulated in a glovebox under nitrogen. A CW laser diode (Coherent OBIS LG 355-20) was used to generate excitation light with an excitation wavelength of 355 nm. In these measurements, pulses were delivered using an acousto-optic modulator (Gooch & Housego, MHP085-6DS2) that was triggered with a pulse generator (WF 1974, NF Co.). The excitation light was focused on a 200 μm beam diameter through a lens and slit, and the excitation power was 2.65 mW. The size and power were checked by using a beam profiler (Thorlabs BP209-VIS) and thermal sensors (Ophir Optronics 3A-PF-12 and StarLite). The emitted light intensity was recorded using a photomultiplier tube (PMT) (Hamamatsu Photonics R928, C3830). The PMT response was monitored on a multichannel oscilloscope (Agilent Technologies DSO5034A).

**Photodegradation measurements**. Thin films were fabricated by spin-coating from 1.2 wt% chloroform solution at 1000 rpm for 60 s on non-fluorescent glass substrates, which were encapsulated in a glovebox. A CW laser diode (NICHIA NDV7375E) was used to generate excitation light with an excitation wavelength of 405 nm. The excitation beam area was 2.5 mm × 2.5 mm circle. The excitation

power was 200 mW for BSBCz-EH. The emission spectra were recorded using spectrometer (Hamamatsu Photonics PMA-12).

**OLED fabrication**. Pre-patterned ITO substrates on a 0.5-mm-thick glass were sonicated for 10 min in deionised water followed by 10 min sonication each in acetone and isopropanol. The substrates were dried with nitrogen before exposing them to a 30 min ultraviolet (UV)-ozone cleaning process. A 30 nm layer of PEDOT:PSS was spin-coated on cleaned ITO substrate followed by annealing at 120 °C for 20 min. BSBCz-EH and mCP-COT solutions were prepared at a con-centration of 7 mg mL$^{-1}$ in chlorobenzene and stirred for 30 min for thorough mixing. Neat and blend layers were spin-coated at 3000 rpm followed by a 100 °C annealing for 10 min to dry out the solvent. TPBi, LiF and Al were evaporated in one go with the help of masks at a pressure below 10$^{-6}$ Torr. Devices were encapsulated and UV treated to avoid degradation in air.

**Transient EL measurements**. Pulse measurements were done with AVTECH Electrosystems Ltd. pulse generator, AV1011B1-B, having a range of 100 ns to 1 ms. The maximum voltage achieved can be 100 V with an ultrafast rise and fall time of 2 ns. A calibrated photomultiplier tube (PMT) was used to collect EL data (Hamamatsu H10721-20) having a 0.57 ns response time. A high-speed current probe UHF711 from Integrated Sensor Technology was used to measure current with a rated response of <0.5 ns. Teledyne LeCroy digital storage oscilloscope (Wavesurfer 900 series), 2 GHz and 10 Gs s$^{-1}$ was used to record pulse data. More details about the setup can be obtained from literature[34].

**Transient absorption spectroscopy (TAS) measurements**. Nanosecond TAS for mCP and mCP-COT was performed in acetonitrile solutions using a broadband pump-probe spectrometer (EOS, Ultrafast Systems, LLC). An Amplified laser system (spitfire ACE, spectra physics) delivering ca. 100 fs laser pulses at 800 nm with a repetition rate of 1 kHz was the excitation source. The laser pulses were coupled to an OPA system (Topas Prime, Light Conversion) to generate "pump" pulses tuned at 330 nm. The samples were prepared by dissolving mCP/mCP-COT in acetonitrile to achieve an optical density of 0.4 at 330 nm in quartz cuvette with the optical path length of 2 mm. A white light continuum 'probe' (ca. 380–900 nm) was generated using a pulsed Nd:YAG based Leukos-STM super continuum light source. The timing of the 'probe' pulses was controlled electronically via the sync trigger from the amplified laser system. The sample solutions were stirred con-tinuously to avoid any degradation during the measurements; absorption spectra were measured before and after the measurements to confirm no degradation due to the pump beam.

## Data availability
The data that support the findings of this study are available from the corresponding author upon reasonable request.

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

## Acknowledgements

We thank the Australian Research Council (ARC DP160100700 and DP200103036), Department of Industry, Innovation and Science (AISRF53765), JST ERATO Grant Number JPMJER1305, JSPS KAKENHI Grant Number JP19H02790, and JSPS Core-to-Core programme for financial support. V.T.N.M. was supported by a UQ International Postgraduate Research Scholarship; V.A. was funded by an Australian Postgraduate Award and A.S. was funded by a UQ's Research and Training Programme. This work was performed in part at the Queensland node of the Australian National Fabrication Facility Queensland Node (ANFF-Q)—a company established under the National Collaborative Research Infrastructure Strategy to provide nano- and micro fabrication facilities for Australia's researchers.

## Author contributions

V.T.N.M. and S.-C.L. conceived the idea of this work. V.T.N.M. developed the syntheses of the triplet quencher and performed material characterisation under the supervision of S.-C.L. V.A fabricated the OLEDs and conducted device simulation under the supervision of J.S. and E.B.N. M.M. supplied the laser dyes and performed the theoretical calculations and photophysical studies on the triplet quencher. T.F performed transient PL under the supervision of M.M. and C.A. A.S. conducted transient absorption spectroscopy experiments under the supervision of E.G.M. and E.B.N. G.K. performed UPS and IPES measurements under the supervision of G.G.A. J.S., C.A., E.B.N. and S.-C.L. verified the methods and data, and supervised the work. All authors contributed to the discussion and analysis of the obtained results, and the input to the final manuscript. V.T.N.M. and V.A. contributed equally to this work.

## Competing interests

The authors declare no competing interests.
