## [Peer Review File · Nature Communications]

Reviewers' Comments:

Reviewer #1:

None

Reviewer #2:

Remarks to the Author:

In this work, the authors report the design, synthesis of a novel solid-state organic triplet quencher based on cyclooctatetraene, and achieve a complete suppression of STA and a 20-fold increase in excited-state photostability of BSBCz-EH under CW excitation. The results are interesting. However, there are some issues must be addressed.

1. The transient absorption decay kinetics analysis is not sufficient, why excited-state absorption bands with maxima at 400 nm and 617 nm could be attributed to the tripled and singlet excited-state absorption, respectively?
2. The transient PL spectra analysis should also be given.
3. Solution absorption (dotted lines) curves in Fig.2a are not clear.
4. There is something wrong with the sentence "both molecules can effectively quench triplet excitons with the same mechanism due to their low triplet energies, in which adi is adiabatic excitation" in Page 11.
5. The authors employ a non-conjugated n-hexyl linker, why? What about influence of the non-conjugated linker length?
6. How about the negative influence of the non-conjugated n-hexyl linker on the electrical properties?

Reviewer #3:

Remarks to the Author:

The manuscript reported a novel triplet quencher mCP-COT with the potential to significantly reduce singlet-triplet annihilation. The authors demonstrated the effectiveness of the triplet quencher by mixing mCP-COT in BSBCZ-EH and showed significantly improved EL and PL characteristics. This is the first report to my knowledge on a successfully designed solid state triplet quencher, which represents a good step towards organic laser diode. I have a few comments on the some wording and technical details of the manuscript as below.

* In the abstract, the wordings "this is the first report of a solid state triplet quencher, exhibiting excellent..." should be revised to "this is the first report of a solid state triplet quencher that exhibits excellent..." or something alike. The reason is because solid state triplet quencher has previously been demonstrated, and the novelty in the current report is in the improved triplet quenching ability.

* Line 261-266 and Figure 4: The data presented in this report is not an apple-to-apple comparison for triplet quenching capability of mCP-COT to ADN in Ref 31. The reason is because baseline (i.e. without triplet quencher) STA is ~ 50% in Ref 31 while baseline STA in the current study is < 25%. The authors should clarify that. Also, by looking at the comparison between Figure 4a and 4b, it seems that the amount of triplet quenching is also dependent on PL pump condition, so does Fig 4b show mCP-COT 20wt% performance is actually worse than Ref 31?

* Line 281-290 and Figure 4: It would be helpful to comment on whether decay in Fig. 4b is temporary or permanent. Also, it is not clear to me why merely changing the pump wavelength from 355 to 405 nm, the transient PL decay rates as well as the triplet quenching amount in Figure 4a and 4b are drastically different. I think the authors should give clear explanation. Is it because BSBCz-EH or mCP-COT has different absorption at different wavelength? Or is it because the pump power densities are very different? Or any other reasons.

* Line 346-350: from fits to EL transients in Fig S10, the authors found STA rate is decreased by >60% as a result of blending mCP-COT in BSBCz-EH. This is a surprise because the function of "triplet quencher" should be to reduce the triplet density, and not to reduce the single-triplet interaction rate. The reason is because triplet-single interaction rate is an intrinsic material property of BSBCz-EH. The authors should explain why STA rate in BSBCz-EH is changed as a result of doing 2% mCP-COT; also, a plot of triplet-density in Figure S10 according to the rate equations should be given to verify that the triplet density is significantly decreased as a result of mCP-COT as triplet quencher.

* Line 328-335 and Figure 5 and Figure S10: There is a small discrepancy in the measured brightness and modeled singlet density, in that @ ~ 100 ns after electrical excitation, EL intensity ratio of blend ($3.5e5$ cd/m²) vs. neat ($0.4e5$ cd/m²) is ~ 9 , while the modeled single density ratio is ~ 6.5 . The authors should explain.

REVIEWER COMMENTS

Reviewer #2 (Remarks to the Author):

In this work, the authors report the design, synthesis of a novel solid-state organic triplet quencher based on cyclooctatetraene, and achieve a complete suppression of STA and a 20-fold increase in excited-state photostability of BSBCz-EH under CW excitation. The results are interesting. However, there are some issues must be addressed.

1. The transient absorption decay kinetics analysis is not sufficient, why excited-state absorption bands with maxima at 400 nm and 617 nm could be attributed to the tripled and singlet excited-state absorption, respectively?

Response: We thank the Reviewer for the valuable suggestion. To further clarify this, we performed nanosecond transient absorption spectroscopy (TAS) for **mCP** and **mCP-COT** in acetonitrile at ambient and deoxygenated (oxygen free) solutions. Furthermore, we added the the Supplementary **Figure S7** to show two distinct transient absorption bands along with comparison of dynamics at 400 nm in **mCP** under ambient and deoxygenated conditions.

We have also made the following changes in the TAS section of the main text as highlighted (pages 9 & 10):

“In order to gain further evidence of triplet quenching, we performed nanosecond transient absorption spectroscopy (TAS) for **mCP** and **mCP-COT** in acetonitrile. In ambient conditions, **mCP** showed long-lived excited-state absorption band with maximum at 400 nm (decay lifetime of 52 ns) and **broad** short lived excited-state feature with maximum at 617 nm (bi-exponential lifetime of 5.8 and 51 ns) (**Fig. 3a, b, S7a**). **Herein, the decay kinetics of the short-lived absorption band (5.8 ns) was found to match closely with the singlet emission decay obtained from TCSPC (see Table S5) measurements (5.3 ns), suggesting this transient absorption band is a result of the singlet excited-state absorption. In order to get further insights into the long-lived feature ($\tau \approx 50$ ns), we performed TAS for **mCP** under deoxygenated conditions. For deoxygenated solution (by using a freeze-pump-thaw method), the lifetime of the long-lived feature increased by more than two orders of magnitude ($\tau \approx 32 \mu\text{s}$) (**Fig. 3c, d, S7b**), suggesting this transient absorption band arises from the triplet excited-states that were otherwise quenched by molecular oxygen under ambient conditions. **Fig. S7d shows the normalised comparison of triplet excited-state absorption decay under ambient and degassed conditions. In case of a deoxygenated **mCP-COT** solution, similar singlet and triplet excited state absorption bands were observed. The decay lifetime of singlet excited-state absorption****

band was found to be 0.9 and 7.3 ns, which is similar to the singlet emission lifetime obtained in the TCSPC measurements (Table S5). Furthermore, triplet excited-state absorption decay of the mCP moiety at 400 nm was found to be significantly quenched ($\tau \approx 26$ ns) (Fig. 3e, f, S7c). The extremely shortened decay lifetime of mCP unit's triplet excited-state absorption in mCP-COT suggests ultra-fast transfer of triplet excitons from the mCP unit to the COT moiety, though due to the extremely low triplet energy level of COT, we could not observe the transient absorption band arising from the COT moiety alone."

Fig. S7: Two distinct transient absorption bands obtained in case of (a) mCP under ambient conditions (aerated), (b) mCP under deoxygenated conditions and (c) mCP-COT (under deoxygenated conditions). **d** Comparison of triplet excited-state absorption decay (at 400 nm) under ambient and deoxygenated conditions in case of mCP.

2. The transient PL spectra analysis should also be given.

Response: We thank the Reviewer for the worthwhile suggestion. We have now added the following TCSPC PL decay curves and tables summarising the fitting parameters in the Supplementary Fig. S6 and Table S5, as well as Fig. S11 and Table S6.

Fig. S6: TCSPC PL decay curves for mCP and mCP-COT in toluene.

Table S5. Summary of photophysical parameters for mCP and mCP-COT in toluene.

	PLQY (%)	Lifetime (ns)	k_r (s ⁻¹)
mCP	43	5.33	8.1×10^7
mCP-COT	6	1.46 (0.014, 0.985, 5.67)	4.1×10^7

Fig. S11: TCSPC PL decay curves for BSBCz-EH neat and blend (with mCP-COT) thin films.

Table S6. Summary of photophysical parameters for BSBCz-EH neat and blend (with mCP-COT) thin films.

	PLQY (%)	Lifetime (ns)	k_r (s^{-1})
Neat	68	1.39	4.9×10^8
1wt% mCP-COT	71	1.56	4.6×10^8
3wt% mCP-COT	71	1.54	4.6×10^8
5wt% mCP-COT	72	1.53	4.7×10^8
10wt% mCP-COT	68	1.43	4.8×10^8
20wt% mCP-COT	67	1.30	5.2×10^8
50wt% mCP-COT	53	1.13	4.7×10^8
90wt% mCP-COT	34	1.10 (0.41, 1.34)	3.1×10^8

3. Solution absorption (dotted lines) curves in Fig.2a are not clear.

Response: We apologise for the confusion. We have now revised the format of Fig. 2a so that: i) solid lines were used instead of dotted lines for the absorption, ii) annotation arrows were added to clearly distinguish absorption and photoluminescence, and iii) the colour scheme of green-blue-red was changed into black-blue-red for **COT**, **mCP** and **mCP-COT**, respectively, iv) the weak absorption of **COT** has been further magnified as an inset in the figure (and as noted there is no **COT** emission). Revised Fig. 2a is shown below and has now been updated on page 7.

Fig. 2: PL Spectra of solution at varying temperatures. a Solution absorption (dotted lines) and normalised photoluminescence (PL, solid lines) spectra of **COT***, **mCP** and **mCP-COT** in toluene (inset shows the weak **COT** absorption). Excitation wavelength = 290 nm.

4. There is something wrong with the sentence “both molecules can effectively quench triplet excitons with the same mechanism due to their low triplet energies, in which adiabatic excitation” in Page 11.

Response: We thank the Review’s comment. We have now revised the description of our computational studies so to improve its comprehension. Specifically, the following changes have now been made to the main text:

- “where the S_2 -ver and S_1 -ver transitions are assigned to HOMO \rightarrow LUMO+1 (57%) and HOMO-2 \rightarrow LUMO (100%)” has now been changed to “where the S_2 -ver and S_1 -ver transitions of **mCP-COT** are mainly populated over mCP moiety [HOMO \rightarrow LUMO+1 (57%)] and COT moiety [HOMO-2 \rightarrow LUMO (100%)]” (page 12).

- Additional explanation of “..., which might be related to the decrease in PLQY for the doped films (vide infra)” has now been added (page 12).
- Additional explanation of “(see S₁-adi and T₁-adi for the adiabatic excitation), and the triplet quenching by COT is known to include non-vertical triplet energy transfer with conformational changes.” has now been added on page 12.
- The explanation of “both molecules can effectively quench triplet excitons with the same mechanism due to their low triplet energies, in which adi is adiabatic excitation” has now been re-written as “it is considered that mCP-COT can also effectively quench triplet excitons with the same mechanism because of its low T₁-adi energy.” (page 12).

5. The authors employ a non-conjugated *n*-hexyl linker, why? What about influence of the non-conjugated linker length?

Response: We thank the Reviewer for the remark. As highlighted in the main text (page 4): a non-conjugated linker is crucial “to maintain the individual electronic properties of mCP and COT” in distinction to the use of a conjugated linker, which would result in issues of potential undesirable changes in the electronic properties of each or both moieties due to conjugation effects.

For this work, in addition to commercially available starting materials for the ease in chemical synthesis as outlined in Fig. 1, the length of the non-conjugated linker was chosen so that the target molecule would achieve a fine balance between (i) solution-processability, and (ii) thermal stability. Specifically, we sought a linker length that is long enough for good solubility of the target chromophore but not too long that may adversely affect the thermal property of the target material such as reduction in its glass transition temperature (for good film stability). Hence, an *n*-hexyl linker was chosen for the study.

To further clarify our choice of the non-conjugated *n*-hexyl linker, we have now revised the statement of “To maintain the individual electronic properties of mCP and COT, we employed a non-conjugated *n*-hexyl linker to give mCP-COT with high solubility in common organic solvents for solution processing.” (page 4) to “To maintain the individual electronic properties of mCP and COT, it is essential that a non-conjugated linker is employed. For mCP-COT to achieve a high solubility in common organic solvents for solution processing without adversely affecting its thermal property, an *n*-hexyl linker was chosen for our initial study”.

Regarding the influence of the non-conjugated linker length, we acknowledge that the effect of varying the linker length is beyond the scope of this work and may be a research topic of future work.

6. How about the negative influence of the non-conjugated n-hexyl linker on the electrical properties?

Response: We thank the Reviewer's remark. *n*-Hexyl group has been a common solubilising moiety employed in multiple high-performing organic semiconductor devices [e.g., poly(3-hexylthiophene), (P3HT) in OFETs/OPVs/OPDs; di-*n*-hexylfluorene oligomers (*Appl. Phys. Lett.* 2017, 110, 023303 or di-*n*-octylfluorene polymers, e.g., 9,9'-dioctylfluorene-co-benzothiadiazole, F8BT, *Adv. Mater.* 2010, 22, 3194) in OLEDs, or as ligand moieties of Ir(III) complexes in phosphorescent OLEDs (*Adv. Mater.* 2002, 14, 581)] that have not shown negative influence on the electrical properties. Our OLED devices using **mCP-COT** (*i.e.*, with a *n*-hexyl linker) have also demonstrated excellent electrical performance with EQEs, reaching theoretical values as shown in the main text (pages 17-21). We have now added a J-V graph of neat and blend OLEDs (shown below) as the new Supplementary **Fig. S15b**, which also shows similar current density characteristics of neat and blend film devices in response to applied voltage. Hence, to the best of our knowledge, the potential negative influence of the *n*-hexyl linker on the electrical properties has not been observed in the context of this work.

Fig. S15b: Compared neat and blend OLED J-V characteristics, showing similar behaviours in the operating voltage range.

We have now added the following statement to the main text (page 20):

“The J-V characteristics are also very similar for neat and blend OLEDs as shown in Fig. S15b, depicting no change in electrical behaviour with the addition of mCP-COT in the operating voltage region (>4 V).”

Reviewer #3 (Remarks to the Author):

The manuscript reported a novel triplet quencher mCP-COT with the potential to significantly reduce singlet-triplet annihilation. The authors demonstrated the effectiveness of the triplet quencher by mixing mCP-COT in BSBCZ-EH and showed significantly improved EL and PL characteristics. This is the first report to my knowledge on a successfully designed solid state triplet quencher, which represents a good step towards organic laser diode. I have a few comment on the some wording and technical details of the manuscript as below.

* In the abstract, the wordings “this is the first report of a solid state triplet quencher, exhibiting excellent...” should be revised to “this is the first report of a solid state triplet quencher that exhibits excellent...” or something alike. The reason is because solid state triplet quencher has previously been demonstrated, and the novelty in the current report is in the improved triplet quenching ability.

Response: We thank the Reviewer for the valid remark. The according text has now been corrected in the Abstract (page 2);

“..., this is the first report of a solid-state triplet quencher that exhibits excellent triplet quenching ability...”

* Line 261-266 and Figure 4: The data presented in this report is not an apple-to-apple comparison for triplet quenching capability of mCP-COT to ADN in Ref 31. The reason is because baseline (i.e. without triplet quencher) STA is ~ 50% in Ref 31 while baseline STA in the current study is < 25%. The authors should clarify that.

Response: We agree with the Reviewer that apple-to-apple comparison with Ref 31 using direct values is not ideal in this case since the two systems have different STA baselines. We have now clarified our comparison between the two triplet quenchers using the **relative magnitude** of STA losses (i.e. by taking the initial STA baseline into consideration) in these two systems. The following changes have now been made:

- In main text, the statement of “were not effective” has now been changed to “were not as effective” (page 14).

- The statement of “In particular, at a 70vol% blend concentration of ADN, the reported PL transient quenching due to STA was still at 17%³¹. In contrast, mCP-COT showed complete suppression of STA at a blending concentration as low as 5wt%.” (page 14) has also been changed to “A comparative analysis of the triplet quenching performance of ADN and mCP-COT was conducted to show the relative drop in the initial STA in the two systems (Fig. S8). The results support the superior triplet management properties of mCP-COT since at the 10wt% concentration mCP-COT removes over 98% STA present originally in the neat system, while the same concentration of ADN results in approximately 25% reduction of STA (Fig. S8)”.
- The new Supplementary Fig. S8 and its legend (including Ref 31, *i.e.*, here Ref S10) have now been added. The Figure numbers of other Figures have been corrected accordingly both in the main text and Supplementary section.

Fig. S8: Relative drop in STA as a function of mCP-COT and ADN quencher concentrations for BSBCz-EH and Alq₃/DCM2¹⁰, respectively. In the absence of STA, there would be no singlet-triplet interaction between populations; therefore, the singlet population should shortly (under 1 μ s) saturate at a steady value where the positive pumping term is balanced out by negative fluorescent ISC and SSA terms (assuming positive contribution of TTA to be negligible) and there is no impact of growing triplet population. Since the singlet population directly correlates to the light intensity, one can treat the difference between peak and steady state in a neat film as a total (*i.e.*, 100%) loss due to STA in a system without triplet quencher. Then, the relative decrease in STA plotted against the quencher concentration can be a rough measure of how successful the triplet manager is in the system.

*Also, by looking at the comparison between Figure 4a and 4b, it seems that the amount of triplet quenching is also dependent on PL pump condition, so does Fig 4b show mCP-COT 20wt% performance is actually worse than Ref 31?

Response: We thank the Reviewer for raising these concerns. Given that Fig. 4b in this work depicts long time photostability (while Fig. 2b from Ref 31 in fact shows microsecond transients instead of long time photostability), it is, therefore, not possible to make any comparison (for this work with Ref 31). Fig. 4a was about the quantification of STA reduction based on transient PL, whereas Fig. 4b was in fact about the photostability of the different films when pumped by a continuous laser source. We apologise for the confusion if the Reviewer have misunderstood these.

To avoid the possible same confusion of Readers likely arisen from our combination of the two figures and our choice of the same colour codes used in Fig. 4a and Fig. 4b, we have now split them into two separate Figures (*i.e.* Fig. 4 and Fig. 5 on pages 15 and 16, respectively) and updated the colour scheme on Fig. 5 (*i.e.* the previous Fig. 4b) to have better contrast, clearly presenting the two different experiments of previous Fig. 4a and Fig. 4b, respectively (where previous Fig. 5 has also been updated as Fig. 6 accordingly).

Fig. 4: Transient PL. Characteristics of encapsulated BSBCz-EH neat film and blend films with different mCP-COT blending concentrations (5wt%, 10wt%, and 20wt%), which can be compared with its blend film with 20wt% in mCP. Excitation wavelength = 355 nm; laser beam excitation power = 2.65 mW; pulse width = 200 µs and pulse interval = 10 ms.

Fig. 5: Photostability study of BSBCz-EH films. PL intensity/initial PL intensity (I/I_0) of a BSBCz-EH neat film (dark green line), a blend film with 20wt% mCP (heart pink line), and a blend film with 20wt% mCP-COT (violet line) measured under CW photoexcitation with a power of 200 mW cm^{-2} at 405 nm. Excitation area = $2.5 \text{ mm} \times 2.5 \text{ mm}$ circle.

* Line 281-290 and Figure 4: It would be helpful to comment on whether decay in Fig. 4b is temporary or permanent. Also, it is not clear to me why merely changing the pump wavelength from 355 to 405 nm, the transient PL decay rates as well as the triplet quenching amount in Figure 4a and 4b are drastically different. I think the authors should give clear explanation. Is it because BSBCz-EH or mCP-COT has different absorption at different wavelength? Or is it because the pump power densities are very different? Or any other reasons.

Response: Many thanks for Reviewer's comments. Indeed, the decay in Fig. 4b (now Fig. 5) is permanent due to photo-degradation. As noted in our previous Response, Fig. 4a and 4b are two separate studies (on the transient PL and photostability, respectively). While the main aim of former Fig 4a (now Fig. 4) is to show the effect of STA in case of neat and blended films through evolution of transient PL signal, which in turn is depiction of evolution of singlet excited-state population, the comparison of long term photostability of neat and blended systems is the focus of former Fig. 4b (now Fig. 5). Fig. 4a (*i.e.* now Fig. 4) shows temporal characteristics, which can be repeated with the same sample since mCP-COT does not have any significant absorption at from 355 to 405 nm, hence only BSBCz-EH is excited in both

the experiments) in contrast to permanent photo-degradation of **BSBCz-EH** films in Fig. 4b (now Fig. 5). Furthermore, as noted by the Reviewer that the pump densities are very different in the two measurements. The temporal characteristics in previous Fig. 4a (now Fig. 4) were performed by using 355 nm excitation source at low excitation densities to avoid photo-degradation while the long term photostability was conducted by using 405 nm excitation as 405 nm excitation source provided higher power densities which are important for photostability study.

* Line 346-350: from fits to EL transients in Fig S10, the authors found STA rate is decreased by >60% as a result of blending mCP-COT in BSBCz-EH. This is a surprise because the function of “triplet quencher” should be to reduce the triplet density, and not to reduce the single-triplet interaction rate. The reason is because triplet-single interaction rate is an intrinsic material property of BSBCz-EH. The authors should explain why STA rate in BSBCz-EH is changed as a result of doing 2% mCP-COT; also, a plot of triplet-density in Figure S10 according to the rate equations should be given to verify that the triplet density is significantly decreased as a result of mCP-COT as triplet quencher.

Response: We agree with the Reviewer that changing the value of k_{STA} during the fit of different samples is not the best way of describing physical processes going in the system (as k_{STA} is an intrinsic material property). In the revised manuscript, we kept the k_{STA} ($4.3 \times 10^{-8} \text{ cm}^3 \text{ s}^{-1}$) constant between neat and blended case while adding another term, $k_{mCP-COT}T_1$, to the triplet rate equation, where the $k_{mCP-COT}T_1$ term is the rate of triplet quenching (depopulation) in the presence of **mCP-COT**. In the neat case its value is kept at 0, while in the blend case the obtained quenching rate of $1 \times 10^{10} \text{ s}^{-1}$ results in significant drop of triplet population, and thus reduction of $k_{STA}S_1T_1$ component in the singlet equation. $k_{mCP-COT}T_1$ has now been added to the triplet rate equation in the Supplementary Information (on page S23) for the blend system.

$$\frac{dn}{dt} = \frac{j(t)}{ed} - k_L n^2$$

$$\frac{dS_1}{dt} = \frac{1}{4} k_L n^2 - k_S S_1 - k_{SS} S_1^2 - k_{ST} S_1 T_1 - 2k_{SP} n S_1 - k_{ISC} S_1 + \frac{1}{4} k_{TT} T_1^2$$

$$\frac{dT_1}{dt} = \frac{3}{4} k_L n^2 - k_T T_1 - \frac{5}{4} k_{TT} T_1^2 - 2k_{TP} n T_1 + k_{ISC} S_1 - k_{mCP-COT} T_1$$

We have also added simulated triplet populations to Fig. S10 (*i.e.* now the new **Fig. S14c**).

We have also modified the discussion on EL simulation in the main text to the following (on page 20);

*In order to confirm STA quenching by mCP-COT in the blend films, rate equations for polaron, singlet and triplet generation were simulated in MATLAB® and the STA rate along with other annihilation rates was extracted from the program. Simulation of neat and blend device EL characteristics from rate equations (see Supplementary Eq. S1) suggests an STA rate (k_{STA}) of $4.3 \times 10^{-8} \text{ cm}^3 \text{ s}^{-1}$ for the neat OLEDs. For blend OLEDs, k_{STA} was kept the same and a new term, $k_{mCP-COT}$, was introduced in the triplet equation depicting contribution of mCP-COT towards rapid triplet depopulation. $k_{mCP-COT}$ was extracted to be $1 \times 10^{10} \text{ s}^{-1}$. **Fig. S14a, b** shows the result of rate equation fitting for the EL response of the neat and blend devices, respectively. It must be noted that the plotted singlet density for both neat and blend devices is for the same current (50 A cm^{-2}) going through both devices. However, singlet density for the blend can be seen as being around eight times the singlet density in neat device (an indication of more STA quenching in neat device). The results of reduced STA quenching indicate the triplet quencher mCP-COT is efficient for the fast triplet decay. **Fig. S14c** gives evidence of triplet populations extracted from neat and blend devices. The triplet population obtained for neat devices is almost 30 times more than that of the blend.*

Fig. S14: Simulated singlet density evolution over time compared with experimental data at 50 A cm⁻². **a** Evolution of singlets in a neat device, actual signal (black) *versus* simulated fit (red); STA rate recovered from simulation was $4.3 \times 10^{-8} \text{ cm}^3 \text{ s}^{-1}$. **b** EL fitting for a 2wt% mCP-COT blend device actual signal (black) vs simulated fit (red) with the same STA rate (as neat is shown). A new parameter $k_{mCP-COT}$ was introduced in the blend case signifying triplet depopulation rate due to COT. $k_{mCP-COT}$ was found to be $1 \times 10^{10} \text{ s}^{-1}$. **c** Evolution of triplet population in neat and blend case is shown depicting efficient triplet recycling in the blend case.

* Line 328-335 and Figure 5 and Figure S10: There is a small discrepancy in the measured brightness and modeled singlet density, in that @ ~ 100 ns after electrical excitation, EL intensity ratio of blend ($3.5 \times 10^5 \text{ cd/m}^2$) vs. neat ($0.4 \times 10^5 \text{ cd/m}^2$) is ~9, while the modeled single density ratio is ~6.5. The authors should explain.

Response: We apologise for the error – many thanks for identifying this. There is an ≈ 8 time increase in EL density/singlet density going from neat to blend devices at 50 A cm^{-2} . The error has now been rectified and can be seen in Supplementary **Fig. S14** (in our Response to previous comment).

Reviewers' Comments:

Reviewer #2:

Remarks to the Author:

The authors have addressed the comments in a satisfactory manner and I am happy to recommend acceptance for publication